# BestCRM: An Exhaustive Search for Optimal Cis-Regulatory Modules in Promoters Accelerated by the Multidimensional Hash Function

**DOI:** 10.3390/ijms25031903

**Published:** 2024-02-05

**Authors:** Igor V. Deyneko

**Affiliations:** K.A. Timiryazev Institute of Plant Physiology RAS, 35 Botanicheskaya Str., Moscow 127276, Russia; igor.deyneko@inbox.ru

**Keywords:** cis-regulatory modules, DNA motifs, transcriptional regulation, promoters

## Abstract

The concept of cis-regulatory modules located in gene promoters represents today’s vision of the organization of gene transcriptional regulation. Such modules are a combination of two or more single, short DNA motifs. The bioinformatic identification of such modules belongs to so-called NP-hard problems with extreme computational complexity, and therefore, simplifications, assumptions, and heuristics are usually deployed to tackle the problem. In practice, this requires, first, many parameters to be set before the search, and second, it leads to the identification of locally optimal results. Here, a novel method is presented, aimed at identifying the cis-regulatory elements in gene promoters based on an exhaustive search of all the feasible modules’ configurations. All required parameters are automatically estimated using positive and negative datasets. To be computationally efficient, the search is accelerated using a multidimensional hash function, allowing the search to complete in a few hours on a regular laptop (for example, a CPU Intel i7, 3.2 GH, 32 Gb RAM). Tests on an established benchmark and real data show better performance of BestCRM compared to the available methods according to several metrics like specificity, sensitivity, AUC, etc. A great practical advantage of the method is its minimum number of input parameters—apart from positive and negative promoters, only a desired level of module presence in promoters is required.

## 1. Introduction

The process of gene regulation involves transcription factors (TFs) binding to specific regions of the genome, such as proximal promoters or distal enhancers, to control gene expression [1]. TF binding to its target sites (TFBS) often occurs cooperatively, forming cis-regulatory modules (CRMs) that are essential for effective and highly specific transcriptional regulation [2]. Collaborations between TFs, whether synergistic or antagonistic, impact gene specificity and regulatory flexibility during processes like tissue development or response to stimuli [3].

The experimental identification of single motifs and CRMs is labor-intensive, leading to the development of computational discovery methods [4]. The basic structures comprising the CRMs are short motifs, which are categorized as structural or sequence motifs [5,6]. The bioinformatic identification of such motifs can be based on sequence statistics [7,8], Chip-seq data [9,10], phylogenetic conservation [11], libraries of known motifs [12,13], and others. Altogether, there are over 150 motif-discovery programs covering a wide spectrum of methodological ideas, types of experimental data, and heuristics (see reviews [14,15,16,17]). 

Today, computational motif discovery has shifted toward identifying entire regulatory modules, termed “composite motifs” or “cis-regulatory modules”. In the literature, both of the terms are used similarly, but in this paper, we will differentiate between modules found based on similarity to experimentally identified modules [18,19] and modules found by evaluating the statistical properties of DNA sequences, for example, overrepresented motif clusters [20,21], which we call CRMs.

Numerous computational algorithms have been developed to identify CRMs by determining potential interactions between TFs based on their co-location [22,23]. Many of these methods, however, require prior knowledge of a TF pair, focus only on statistically overrepresented single motifs, and need pre-defined parameters like thresholds for position weight matrices (PWMs) [21,24,25]. Recent efforts aimed to overcome these limitations by employing various strategies, such as searching for clusters of binding sites, comparing functional conservation between species, and applying complex statistical methods like the hypergeometric test [20,21,26,27].

The general challenge of module discovery involves inferring both the model representing a module and identifying its locations on sequences. Most methods require users to provide candidate single motif models, typically in the form of IUPAC consensus strings or PWMs [28]. Some, like Co-Bind [29], LOGOS [30], and CisModule [31], infer the module’s structure and location without stringent input requirements. Others, like Hexdiff [32], necessitate known CRMs for training. The definition of what a significant combination is varies across methods—MSCAN [33] searches for regions with unusually high densities of binding sites, and ModuleSearcher [34] and CREME [35] look for specific combinations of motifs that co-occur in regulatory regions of related genes. 

Efforts to improve existing methods involve approaches like Fuzzy Clustering [20], Co-occurring Pattern Search (COPS) [21], and randomized occurrence frequency [26]. However, these approaches may face limitations in running time and memory consumption, making them less applicable to large and complex genomes. Additionally, some methods, such as those by Nandi et al. [26] and Hu et al. [27], require many user-specified (and mostly unknown) parameters, potentially impacting their performance. One notable method, MatrixCatch [36], focuses on experimentally verified regulatory elements and recognizes composite elements similar to those stored in the TRANSCompel database [19]. While MatrixCatch outperforms statistical methods, it has limitations in detecting novel TF pairs. 

The presented method aimed to target the following several challenges: detect novel motif combinations; minimize the number of input parameters from the user by estimating them using input DNA sequences; perform an exhaustive search to find the most significant CRMs; and finally, be computationally effective. The exhaustive search (also called brute force search) is the examination of all combinations of motif types, thresholds, and distances between motifs. The computational efficacy is achieved using the special hash function to organize the space of potential modules, followed by a reduction in search space using limitations on the module’s presence in datasets. Schematically, the method is represented in Figure 1.

## 2. Results

Testing of performance is a critical step, describing the general advantages of a method and also specific cases where a method can be of the most use. The following principles were suggested in [37]: testing should use the same datasets and metrics as previous methods so the results are directly comparable; the selection of comparative methods should be objective and the number of methods substantial; and finally, self-made datasets and user-defined performance metrics are used only in addition to the above to show a method’s superiority in specific applications. Accordingly, performance testing was performed using the established datasets [5] containing known CRMs as targets and a dataset of tissue-specific promoters [38] with unknown CRMs. 

### 2.1. Performance of the Method on Recognition of Known CRMs 

For the evaluation of BestCRM performance, we utilized an established benchmark dataset [39] and our previous program MatrixCatch [36]. As a metric of quality, we opted for the nucleotide-level correlation coefficient (nCC) rather than binary true/false predictions, upon which measures like true positives (TPs), false negatives (FNs), etc., are calculated. This choice was made because the values of true/false predictions did not effectively address scenarios where a predicted module only slightly overlaps with a real one or is considerably longer than a real one [39]. For single, short motifs, it can be accepted when a slight overlap is counted as a true prediction, but for longer CRMs, it will mislead the search. For example, MCAST [40] identified in the 500 bp breast dataset [36] a module of 355 bp long with 23 motifs as a top hit. Clearly, programs predicting very long modules (hundreds of nucleotides) will have higher chances to make “correct” predictions just by chance if a slight overlap is counted as a true hit. In contrast, nCC assesses how many nucleotides were actually predicted correctly, and on this basis. it calculates measures like sensitivity (Sn), specificity (Sp), positive predictive value (PPV), and others. 

The chosen benchmark [39] comprises known CRMs together with PWMs and promoter sequences associated with them. This benchmark consists of several datasets with true PWMs along with additional “noise” matrices. The higher the noise, the more difficult it is to identify the true modules. BestCRM was executed with its default parameters on all datasets, and the results were submitted for evaluation, as described in [39]. Additionally, we included our previous program MatrixCatch, designed specifically to detect known CRMs, in the comparison. 

The results are presented in Figure 2 (Figure 2B,C depicts the 75% noise level, and data for all noise levels are depicted in Appendix A in the Supplementary). BestCRMs outperforms other statistical methods like CMA, ModuleSearcher, Stubb, MSCAN, MCAST, Cister, and Cluster-Buster, for all noise levels according to nCC and other performance characteristics like Sn SP, PPV, AUC, etc., except for ModuleSearcher at the point “99% noise” (Figure 2A,B). One of the important properties of BestCRM is its stable performance over different classes of TFs, while most other methods show a strong bias in performance toward particular classes of TFs. For example, a pair IRF-NFkB is reliably detected by all the methods, while Ebox-Ets is quite difficult to detect (Figure 2C). Interestingly, for Cister, the module Ebox-Ets is the second-best at recognition. BestCRM and MatrixCatch show more uniform performance over different classes of transcription factors. 

BestCRM constantly performs above all the statistical methods and slightly below MatrixCatch. The reason for that is the distinct nature of these methods (de novo identification and library-based), which require a detailed explanation of this point. MatrixCatch uses a library of known CRMs based on the TRANSCOMPEL database [19], and the test dataset also uses the same database together with the original data collected by the authors [39]. Obviously, MatrixCatch has an advantage over methods utilizing only the statistical properties of DNA sequences. The quality of CRM identification in non-annotated DNA sequences will be discussed in the next section. 

### 2.2. Tests on Recognition of Novel CRMs 

An experimental investigation of gene expression across diverse human tissues revealed significant variations in transcription, including transcriptions initiated by alternative promoters [38], making the identified tissue-specific promoters (i.e., promoters driving similar expression patterns) ideal candidates for bioinformatics analysis. Our objective is to explore the presence of potential regulatory modules using several approaches. 

To uncover CRMs, we selected the methods used above and one new method, PC-Traff [41]. Among eight programs, two (MSCAN and Stubb) were unavailable; Cluster-Buster and Cister could not be applied due to other input sequence requirements. MCAST identified lengthy modules with numerous motifs (as mentioned above, a module of 355 bp long with 23 motifs), which, from a biological perspective, has no meaning. PC-Traff identified too many modules with highly similar motifs (the problem of self-correlated motifs is discussed later) and was also excluded. Ultimately, only five programs—BestCRM, MatrixCatch, CisModule, ModuleSearcher, and CMA—were employed for the analysis.

All of the programs were run with default parameters and adjustments: CMA, ModuleSearcher, and CisModule, with a maximum of two motifs per CRM, and in ModuleSearcher, the “Number of top scoring modules to return” was set to 10. CMA was set to output five CRMs (maximum allowed) and optimize the distance of a module. The resulting modules were optimized in order to maximize the ratio $C_{CRM}^{+}/C_{CRM}^{-}$ (these and the following values are introduced in the Material and Methods section), provided that the boundary conditions $C_{CRM}^{+}\geq C_{min}^{+}$, and $C_{CRM}^{-}\leq C_{max}^{-}$ hold true, by varying thresholds for both PWMs. The CRM with the highest ratio is reported as a top hit for each program. Various sets of boundary conditions $\left(C_{min}^{+},C_{max}^{-}\right)$ were used: (0.90, 0.50), (0.75, 0.50), (0.66, 0.50), (0.50, 0.25), and (0.33, 0.15), which can be interpreted as a search for regulatory modules that are common to most of the promoters or to a small subgroup.

The results are presented in Table 1 and Appendix A in Supplementary. As it can be seen, in each specificity group, BestCRM found modules in more datasets compared to other methods. For example, in the top group ($C_{CRM}^{+}\geq 0.90$ and $C_{CRM}^{-}\leq 0.5$0), BestCRM found regulatory modules in breast- and prostate-specific promoters, while MatrixCatch only found regulatory modules in prostate-specific promoters. Other programs failed to identify any CRMs with such stringent criteria. 

The CRM found in the breast dataset consists of motifs for interferon-regulatory factors (IRFs) and nuclear factor-κB (NFKB), found by respective PWMs with the thresholds 0.92 and 0.68, respectively, and located at a maximum distance of 130 bp (Figure 3). The family of IRF plays an important role in defense against intracellular pathogens, including the expression of intrinsic anti-microbial defense and the production of interleukin-12 (IL12), which are essential for the priming of early T cell-mediated immune responses [42]. Second, TF-NFκB consists of a family of transcription factors that play a critical role in inflammation, immunity, cell proliferation, and survival [43]. So, based on our statistical analysis of motif combinations in promoters active in breast tissue, a novel potential CRM was found with functionally relevant transcription factors. 

The second CRM, found by both BestCRM and MatrixCatch in prostate-specific promoters, is a known module composed of two copies of C/EBP motifs located at a distance of up to 35 nucleotides and bound by C/EBP-related proteins [44]. C/EBP transcription factors were found to upregulate metastatic gene expression in human prostate cancer cells [45,46], which demonstrates that BestCRM without any prior knowledge is able to identify regulatory modules, the functionality of which has been confirmed by several independent studies. MatrixCatch found this CRM because it was in the library used for the search. Other programs could not identify any modules in the prostate dataset. 

The above example clearly demonstrates the differences between BestCRM and MatrixCatch—the later uses the library of known regulatory modules and detects new instances of these modules using a complex measure of similarity. BestCRM makes an exhaustive search of all the possible combinations of single motifs located at different distances and, therefore, is able to reveal novel combinations based on the statistics of motif presence in positive versus negative promoters. 

Altogether, MatrixCatch is able to accurately identify the regulatory modules and shows a slightly higher recognition performance (Figure 2), but it is limited to the known modules. In contrast, BestCRM can identify new combinations with a recognition power comparable to MatrixCatch, and it is much more precise compared to other statistical methods. 

The different nature of the two algorithms also explains why MatrixCatch reports so many modules (Appendix A in Supplementary). For example, in breast datasets, MatrixCatch reports 67 CRMs in the group ($C_{CRM}^{+}\geq 0.66$ and $C_{CRM}^{-}\leq 0.5$0) because it outputs each match between the sequence and every example in the library. Assuming that some CRMs have dozens of examples, the output from MatrixCatch can be quite exhaustive. BestCRM searches for the most optimal configuration for each type of CRM, and therefore, the output is more concise. 

## 3. Discussion

The exploration of gene transcriptional regulation through bioinformatic techniques is a common practice in biomedical research, and the method presented here significantly contributes to this field. The program BestCRM is aimed at the discovery of cis-regulatory modules in gene promoters and enhancers, and it is equipped with several motif libraries, test datasets, and examples. The program imposes no restrictions on the size and number of the promoters/enhancers, and it is suitable for analyzing small and short DNA loci or extensive datasets representing entire genomes. The stringency of the search can be easily adjusted using a minimal and maximal presence of CRMs, which has a clear biological interpretation of transcriptional co-regulation. Testing according to the guidelines for objective comparison [37], including the recognition of known CRMs and comparison with other programs using established datasets, showed that BestCRM consistently outperforms alternative methods. In a specific study focusing on tissue-specific promoters, the program successfully pinpointed a candidate regulatory module unique to breast- and prostate-active promoters, offering a promising avenue for further investigation. In contrast, alternative methods exhibited lower specificity, with some failing to identify CRMs in promoters for many tissues.

The exhaustive search deployed in BestCRM represents a class of computational problems with NP complexity, with computational time growing exponentially with the number of parameters. Usually, one would require a kind of heuristic or simplification to speed up computations, which always leads to finding only local optima. Here, we showed that by constructing a special hash function and reducing the search space, it is possible to verify all combinations of parameters with a dimensionality of three. Our investigations show that it is feasible to increase the dimensionality up to four and maintain a still-acceptable computational time. For example, additional parameter such as a minimal distance, composite score, or a third motif can be introduced. The composite score is the most promising as it represents cooperativity between protein factors, which is observed in real modules [36]. This effect consists in stabilizing the binding of the second protein factor on a weaker DNA motif (also called co-motif) via protein–protein interactions with a first factor, thus, forming a stable dimer–DNA complex. This idea was realized in MatrixCatch, and it can be added to BestCRM, but this would require some additional optimizations for use in modern multicore and multiprocessor systems to achieve effective parallel computations. Another promising direction for further development is the use of different maximization functions. For example, the presence of positive promoters $C_{CRM}^{+}$ can be maximized rather than the ratio $C_{CRM}^{+}/C_{CRM}^{-}$. Coupling the optimization with numerical gene expression values can also refocus the search to give more weight to genes with high levels of expression.

Another very important feature of the new method is its tolerance to the correlated PWMs. The libraries of motifs often contain similar PWMs, for example, describing different subsets of known motifs sorted by experimental evidence. These highly similar PWMs have different names, and they are accepted by search programs as unrelated. This results in identifying artificially coupled motif pairs, which have nothing to do with biological co-regulation or collaboration between respective transcription factors. Another issue is when two unrelated transcription factors indeed share similar DNA binding motifs. This is often when the protein factors actually compete for the same binding motif, displacing each other on a DNA sequence, but this contradicts the concept of CRM. This problem is exemplified by the PC-TraFF [41] program applied to the breast dataset. The program outputs a number of significant modules, and the top five of them consist of fully correlated motifs (Appendix A in Supplementary). Indeed, the top module for TFs PU1 and ETS consists of motifs GAGGAAG and CTTCCTC, which are exact reverse complements to each other. The next four modules are similarly artificially correlated (Appendix A in Supplementary).

The concept of contrasting positive and negative sets realized in BestCRM automatically solves the described problem—correlated PWMs stay correlated in any sequence, and their presence in negative promoters will be above the allowed maximum $C_{max}^{-}$. This removes the need for removing redundant PWMs, clustering similar motifs, adopting specific codon usage [47], or post-filtering results, which has been realized in some methods. BestCRM only outputs motif pairs that are specific to the dataset of interest.

For practical investigations of gene promoters, the following workflow could be suggested: First, with a set of positive promoters of co-regulated genes, programs like MatrixCatch are best to be applied to find out if there are any known regulatory modules. Modules similar to the known ones will give immediate biological explanations of the possible regulation mechanisms. Next, statistical approaches like BestCRM can be applied that will show if promoters share common modules. These modules can then be mapped to databases to obtain a functional view of the possible mechanisms of regulation.

## 4. Materials and Methods

### 4.1. Basic Definitions

A CRM consists of a pair of single DNA motifs defined by two PWMs (PWM1, PWM2) with respective thresholds (Thr1, Thr2) for a minimal PWM score, and it is located at maximum distance D (graphically represented in Appendix A in Supplementary). Let S^+^ be a set of promoter sequences with a positive response in some experiment and S^−^ with no or negative response; n^+/−^ is a number of sequences in S^+/−^. We say that a CRM has a presence $C_{CRM}^{+}$ on S^+^ if it can be identified on $n^{+}\cdot C_{CRM}^{+}$ sequences. In other words, $C_{CRM}^{+}$ is a share of sequences that contain at least one module.

### 4.2. Problem Definition

By varying the types of single motifs, thresholds and distance a combination should be found so that the presence of the CRM is above some minimal value on positive sequences, below some value on negative, and the ratio between these values is maximized. Formally, it looks as follows: $C_{CRM}^{+}\geq C_{min}^{+}$, $C_{CRM}^{-}\leq C_{max}^{-}$, $C_{CRM}^{+}/C_{CRM}^{-}\to max$, where $C_{min}^{+}\;and\;C_{max}^{-}$ are the boundary conditions (by default, we set these values to $C_{min}^{+}=0.75$ and $C_{max}^{-}=0.5$).

### 4.3. Accelerating by the Hash Function

By default, the ratio $C_{CRM}^{+}/C_{CRM}^{-}$ should be calculated for all combinations of Thr1, Thr2, and D, and for all pairs of motifs, which represent a class of so-called NP-hard problems, and it is computationally very intensive. Fortunately, this procedure can be optimized using the following idea: Let us assume a hash function H (Thr1, Thr2, D) that shows how many sequences in our set contain at least one CRM found with parameters (Thr1, Thr2, D). The major property of this function is that it is monotonically increasing with the decrease in the PWM thresholds and the increase in D. Indeed, if a CRM is found with a Thr1 = 0.9, then it will also be found with a Thr1 = 0.89, 0.88, 087, and so on (same for Thr2). Reducing the thresholds will only increase the number of CRMs and, hence, the number of promoters with them. The same is valid for the length D. Normalizing H with the number of promoters in a set will give the presence value *C_CRM_*. Further we will always use the normalized H.

Therefore, starting from the point (Thr1 = 1.0, Thr2 = 1.0, D = 0) and consecutively decreasing thresholds, one needs to find a point where $C_{CRM}^{+}\geq C_{min}^{+}$. From this point, one would need to control how many modules the negative promoters contain (if $C_{CRM}^{-}\leq C_{max}^{-}$), when further decreasing the thresholds. The area where both conditions are met defines the parameter range of the possible solutions.

Figure 4 gives an example of the above idea using prokaryotic PWMs for TFs REX and FUR from the PRODORIC database [48] and *Bacillus subtilis* promoters as a positive set versus random genomic sequences as a negative (all datasets are given on the program website). So, at the point (Thr1 = 1.0, Thr2 = 1.0), neither *B. subtilis* promoters nor random sequences contain a CRM. By reducing the thresholds more and more, promoters contain the module, and at values (Thr1 = 0.65, Thr2 = 0.8) more than 0.75 ($C_{min}^{+}$) of the positive set contain the CRM. This indicates that from this point and toward lower thresholds, positive promoters will always have a higher number of CRMs, which defines the area of possible values for Thr1 and Thr2 (red mesh at 0.75 ($C_{min}^{+}$) Figure 4). After further decreasing the thresholds, the number of random sequences containing the CRM increases, reaching the limit of 0.5 at Thr1 = 0.425 and Thr2 = 0.7, which in turn, defines the area of possible values based on the negative set (red mesh at 0.5 ($C_{max}^{-}$), Figure 4). The overlap of those areas defines the parameter range where the ratio $C_{CRM}^{+}/C_{CRM}^{-}$ should be maximized, which is only a tiny part of all the possible parameter ranges (red area on the bottom, Figure 4). Therefore, using the described idea, one can significantly increase computational efficiency and make an exhaustive search practically feasible on a regular PC (as used in this work—CPU Intel i7, 3.2 GH, 32 Gb RAM). Of note, here, the parameter distance between motifs D is not assumed to simplify explanations and make it possible to draw the surfaces in 3D, but in the program, the calculations are performed for three parameters: Thr1, Thr2, and D.

Overall, the entire algorithm consists of several steps and is repeated for each possible motif pair: dynamically build a hash function for the positive set and define the area where the boundary constraint $C_{CRM}^{+}\geq C_{min}^{+}$ is met; then, on that area, build the function for the negative set controlling $C_{CRM}^{-}\leq C_{max}^{-}$, and finally, maximize the ratio $C_{CRM}^{+}/C_{CRM}^{-}$ for all valid parameter combinations.

More detailed descriptions with formulas are given in the Supplementary including another example of the hash function for prokaryotic transcription factors (Appendix A).

### 4.4. Datasets Used for Performance Testing

The method was tested according to the principles suggested in [37]. Accordingly, the performance testing was performed using the established datasets [5] and a dataset of tissue-specific promoters [38]. The first dataset [5] contains ten sets of DNA promoters with CRMs of different types, each set varying from 7 to 16 promoters. To simulate different complexity levels, the dataset contains 60 sets of Transfac PWMs with different ratios of PWMs related to the target CRMs and “noise” PWMs. An evaluation of the prediction results was performed remotely on the website [5].

The second dataset contains promoters driving specific gene expression in 11 human tissues [39]. A total of 22 datasets of positive and negative promoters of lengths 500 bp and 1 kb, covering regions (−400 to +100) and (−900 to +100) around the Transcription Start Site (TSS), respectively, were generated (all datasets are available on the program website). The number of sequences ranges from 17 for prostate-specific promoters to 761 for cerebellum-specific promoters, with an average of 184 promoters per set.

## 5. Conclusions

Here, we have presented a novel method, BestCRM, for the identification of cis-regulatory modules in gene promoters and enhancers. The software is supplied with several PWM libraries and can be easily extended by the user. The performance of the method is shown on the established benchmark data and on the real task of promoter analysis of co-expressed genes. Implementation of the exhaustive search through the entire parameter space guarantees the identification of globally optimal CRM once it exists. The program source code executables for Windows and Linux, together with all the required libraries and examples, are freely available at https://github.com/Deyneko/BestCRM (accessed on 25 January 2024).

## Figures and Tables

**Figure 1 ijms-25-01903-f001:**
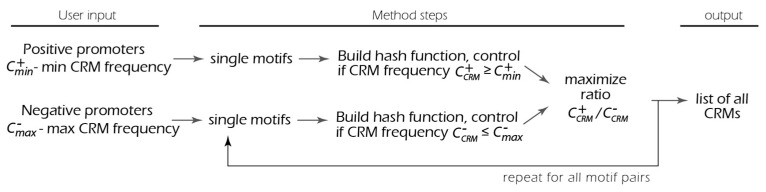
Schematic representation of the method. Exhaustive search (also called brute-force search) is a simple enumeration of all the possible motif pairs. Acceleration using the four-dimensional hash function allows the search to be completed in several hours.

**Figure 2 ijms-25-01903-f002:**
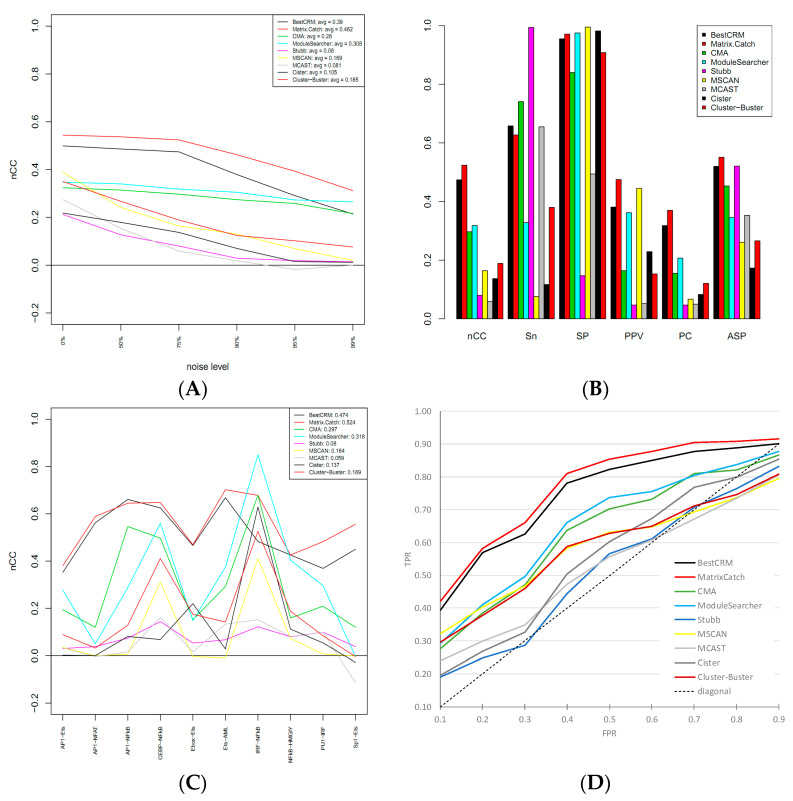
Comparative testing of BestCRM on the dataset from [38]. (**A**). Performance over different noise levels. nCC (nucleotide correlation coefficient) shows how many nucleotides within CRM on average were predicted correctly. (**B**). Different parameters reflect the quality of predictions: Sn—sensitivity; SP—specificity; PPV—positive predictive value; PC—performance coefficient; ASP—average site performance. (**C**). nCC values over different classes of transcription factors. (**D**). Receiver operating characteristic (ROC) and area under curve (AUC) of recognition methods. (**A**–**C**) produced by evaluation software [39].

**Figure 3 ijms-25-01903-f003:**
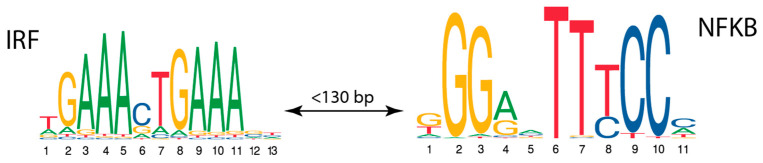
Novel CRM identified in promoters of human genes with specific expression in breast (dataset from [38]). CRM consists of motifs for transcription factors IRF and NFKB located at a maximum distance of 130 bp.

**Figure 4 ijms-25-01903-f004:**
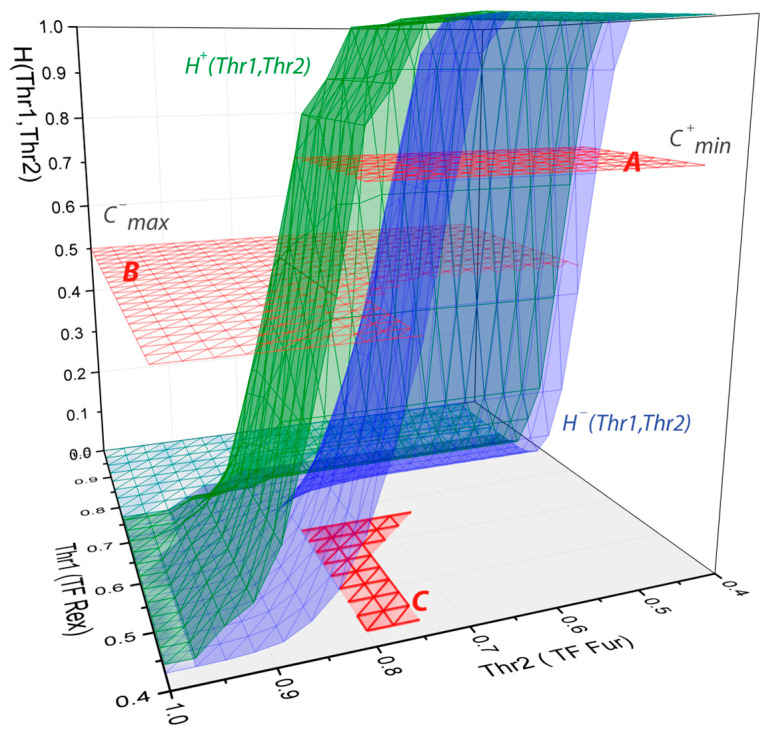
Hash functions for transcription factors REX and FUR. Green surface (H^+^) is a hash function built for positive promoters (*B.subtilis* promoters), and a blue surface (H^−^) is for the negative (random genomic sequences). A—area of parameters (Thr1, Thr2), where more than $C_{min}^{+}$ of positive promoters contain CRM; B—area of parameters (Thr1, Thr2), where less than $C_{max}^{-}$ of negative promoters contain CRM; C—intersection of A and B area, where the ratio $C_{CRM}^{+}/C_{CRM}^{-}$ should be maximized. Parameter D—distance between motifs is not presented here to be able to draw the surfaces in three-dimensional space.

**Table 1 ijms-25-01903-t001:** Recognition of regulatory modules in tissue-specific promoters. The values represent the number of datasets in which respective programs found at least one CRM with the required level of presence in positive $\left(C_{min}^{+}\right)$ and negative $\left(C_{max}^{-}\right)$ promoters. BestCRM identifies CRMs in more datasets for all settings of motif presence.

	$\mathrm{Boundary}\;\mathrm{Conditions}:\;(C_{min}^{+}/C_{max}^{-})$				
	0.90/0.50	0.75/0.50	0.66/0.50	0.50/0.25	0.33/0.15
BestCRM	2	6	7	5	7
MatrixCatch	1	4	6	4	5
CMA	0	1	3	0	1
ModuleSearcher	0	1	6	1	3
CisModule	0	0	1	1	2

## Data Availability

The program source code and executables for Windows and Linux together with all required libraries and examples are freely available at https://github.com/Deyneko/BestCRM (accessed on 25 January 2024).

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
