# Peer review of "BestCRM: An Exhaustive Search for Optimal Cis-Regulatory Modules in Promoters Accelerated by the Multidimensional Hash Function"

_ijms, 2024, doi:10.3390/ijms25031903_

Round 1
Reviewer 1 Report
Comments and Suggestions for Authors
What do you mean by a regular laptop? mention the configuration in the abstract.
Mention the performance metrics in the abstract
Line 32-33: elaborate on computational motif discovery methods and consider citing "Sahu TK, Rao AR, Vasisht S, Singh N and Singh UP(2012), Computational Approaches, Databases and Tools for in silico Motif Discovery, Interdisciplinary Sciences: Computational Life Sciences, 4(4):239-255"
You have mentioned 10 tools for module discovery from line 49-64, but why you have compared your tool with only few.
Line no 76 insert "of" between "recognition" and "CRM"
Line no 77 to line no 97 seem like the part of methodology. Shift it to appropriate place in methodology.
add reference to line no 81-83
Line no 100-what are the other statistical methods, name them
Figure 1 poor figure quality. Improve resolution
Include AUC-ROC in your performance metrics
Line 118 "test dataset is also uses" remove "is" from the statement
Give statistics of all the datasets used in the study in detail with the number of data points in each dataset.
Line 124-127 should be a part of discussion
Lines 129 to 149 should be the part of method
Line no 150 "As can be seen" to "As it can be seen"
Line no 156. Expand IRF and NFKB at their first appearance
Result should be confined to performance of the method and comparison details
Line no 178-194 should be a part of discussion.
Mention in the legend that what the numbers in the table 1 indicate
As author claim that his algorithm is time efficient, he should compare the existing methods with his method based on time in a computer(configuration to be mentioned).
Give appropriate performance metrics rather than reporting the numbers.
What do you mean by the 11 number of datasets. Elaborate datasets with data points in detail in the material and method section.
Re-configure the materials and methodology section with the suggested changes and an additional section for datasets.
Why the default values are set to 0.75 and 0.5
Define H+CRM and H-CRM properly
Line no 302: "That indicate that" to "it indicates that"
Add a flow chart of you algorithm in methodology section
309-310 write the configuration of the regular PC you m3eant
Discussion is poorly written without sufficient corroboration to earlier works. Improve the discussion with suggested changes give in previous comments
Comments on the Quality of English Language
Carefully check the English and grammar throughout the manuscript.
Author Response
Reviewer 1
I thank all of the reviewers for constructive suggestions, following is the replies. All corrections of the English are accepted as suggested. Please note, not all corrections are marked in red, there are many more small changes not marked throughout the text.
- What do you mean by a regular laptop? mention the configuration in the abstract.
Here I meant not a multiprocessor high end servers. Added “… regular laptop (as for example, CPU Intel i7, 3.2GH, 32Gb RAM).”
- Mention the performance metrics in the abstract
Added “… methods according several metrics like specificity, sensitivity, AUC etc.”
- Line 32-33: elaborate on computational motif discovery methods and consider citing "Sahu TK, Rao AR, Vasisht S, Singh N and Singh UP(2012), Computational Approaches, Databases and Tools for in silico Motif Discovery, Interdisciplinary Sciences: Computational Life Sciences, 4(4):239-255"
Citation inserted, plus new paragraph on motif discovery: 2nd paragraph.
- You have mentioned 10 tools for module discovery from line 49-64, but why you have compared your tool with only few.
Not all programs are available: “Among eight programs, two (MSCAN and Stubb) are unavailable, Cluster-Buster and Cister couldn't be applied due to other input sequence requirements.”
+Line no 76 insert "of" between "recognition" and "CRM" - done
- Line no 77 to line no 97 seem like the part of methodology. Shift it to appropriate place in methodology.
Methodology in my case is the description of the method, here it is about testing on a specific data. Other data will require other metrics, as in the next section on searching unknown CRMs. There TP, FN etc. cannot be calculated.
- add reference to line no 81-83
Added :[ Klepper 2008]
- Line no 100-what are the other statistical methods, name them
These are on figure 1, now in better resolution.
- Figure 1 poor figure quality. Improve resolution
I'm very sorry for that. As a reviewer I also receive sometimes documents with images in low quality. But now I know what is the problem - the software on mdpi website reduces the quality too strong when converting from docx to PDF. Now I have converted myself retaining original quality. Ps. Fig 1 are original image produced by evaluation software, I would like to keep it original as is.
- Include AUC-ROC in your performance metrics
Included in Fig.1.
+ Line 118 "test dataset is also uses" remove "is" from the statement - done
- Give statistics of all the datasets used in the study in detail with the number of data points in each dataset.
Included a subsection on datasets to M&M. lines 331-
+Line no 150 "As can be seen" to "As it can be seen"
+Line no 156. Expand IRF and NFKB at their first appearance - done
- Line no 178-194 should be a part of discussion.
Yes, fits better to discussion, moved there.
-Mention in the legend that what the numbers in the table 1 indicate
Legend redone.
- As author claim that his algorithm is time efficient, he should compare the existing methods with his method based on time in a computer(configuration to be mentioned).
It is not time efficient to existing methods, it is efficient compared to simple exhaustive search without optimization, which will take years. That is why it is not realized as a program. With optimization the time is already acceptable, but still is several hours.
- Give appropriate performance metrics rather than reporting the numbers.
Now ROC is added to Figure 1.
- What do you mean by the 11 number of datasets. Elaborate datasets with data points in detail in the material and method section.
11 is a total number of datasets, now in description of datasets added to M&M.
- Why the default values are set to 0.75 and 0.5
It is from our experience; more stringent values will produce no CRMs in many datasets we investigated.
- Define H+CRM and H-CRM properly
Added explanations to fig 4 and text.
+Line no 302: "That indicate that" to "it indicates that" -- done
- Add a flow chart of you algorithm in methodology section
Added at the beginning. New figure 1.
-309-310 write the configuration of the regular PC you m3eant
Added.
- Discussion is poorly written without sufficient corroboration to earlier works. Improve the discussion with suggested changes give in previous comments
Discussion redone.
+Carefully check the English and grammar throughout the manuscript. – done
Reviewer 2 Report
Comments and Suggestions for Authors
In manuscript “BestCRM: exhaustive search for optimal Cis-Regulatory Modules in promoters accelerated by multidimensional hash function.”, Igor V. Deyneko have proposed a novel computational model, namely BestCRM, for identifying cis-regulatory modules. An exhaustive approach, while the NP-complexity is tackled to some extent, was utilized to solve the problem effectively. Overall, the work shows promise, and with the suggested revisions, I believe it has the potential to make a valuable contribution to the field.
Major issues:
1. Figure 1 is super blurred, where I cannot read any texts from the figure. Please replace with a higher quality figure. Due to the quality issue, I cannot tell which bar and line represents which method.
2. It would be better to define and explain terms, such as
and
,, in the chapter 2.2.
3. What are Thr1, Thr2 in chapter 4.1?
Minor issues:
1. For the TF motifs of IRF and NFKB, which gene does the promoter belongs to? Is there any literature supporting these two TFs act on that gene?
2. Please add y labels for figure S3.b.
Author Response
Reviewer 2
In manuscript “BestCRM: exhaustive search for optimal Cis-Regulatory Modules in promoters accelerated by multidimensional hash function.”, Igor V. Deyneko have proposed a novel computational model, namely BestCRM, for identifying cis-regulatory modules. An exhaustive approach, while the NP-complexity is tackled to some extent, was utilized to solve the problem effectively. Overall, the work shows promise, and with the suggested revisions, I believe it has the potential to make a valuable contribution to the field.
Major issues:
- Figure 1 is super blurred, where I cannot read any texts from the figure. Please replace with a higher quality figure. Due to the quality issue, I cannot tell which bar and line represents which method.
- I'm very sorry for that. As a reviewer I also receive sometimes documents with images in low quality. But now I know what is the problem - the software on mdpi website reduces the quality too strong when converting from docx to PDF. Now I have converted myself retaining original quality. Ps. Fig 1 are original image produced by evaluation software, I would like to keep it original as is.
- It would be better to define and explain terms, such as and ,, in the chapter 2.2.
This is introduced later in M&M section, such structure was suggested by mdpi. I added to the text “…(these and the following values are introduced in material and methods section)”
- What are Thr1, Thr2 in chapter 4.1?
This is a for minimal PWM score, if reached a sequence is recognized as a motif. Now changed to:
“…single DNA motifs defined by two PWMs (PWM1, PWM2) with respective thresholds (Thr1, Thr2) for minimal PWM score,”
Minor issues:
- For the TF motifs of IRF and NFKB, which gene does the promoter belongs to? Is there any literature supporting these two TFs act on that gene?
This CRM was found in 19 promoters of genes specifically expressed in breast human tissue. I discuss a little on that “TF – NFκB, consists of a family of transcription factors with a critical role in inflammation”, that could be a kind of support. If one or two genes are reported in literature to be regulated by that TFs, that would be of little help. It should be a statistically sound number of genes.
- Please add y labels for figure S3.b.
All performance measures have values from 0 to 1 (y-axis). Some images overlapped, now reformatted.
Reviewer 3 Report
Comments and Suggestions for Authors
Title: How does the author name the algorithm "BestCRM", if in future some better algorithm will be developed then how this "Best" will be justified? The name of the algorithm should be changed.
Graphical unit interface or URL links should be given to show how this algorithm looks and works.
Picture of the BestCRM should be given as a Figure or links to allow users to interact with it.
Introduction: Providing some examples of related computational algorithms with online links will be useful for the reader.
Figure 1: The graphics of the figure are not clear, and no legends are visible.
From where the sequence data was collected to test the algorithm?
How does the author say the algorithm works best only based on identifying the promotor with minimum input parameters? More input parameters will usually give the best output.
Is BestCRM only used for Humans? Can this method identify the CRM of prokaryotes (bacteria/archaea)? If it is, then how is it distinguished from different eukaryotes?
Does BestCRM reveal the specific patterns in different Extremophiles or Archaea? Then, cite the following paper: https://www.nature.com/articles/s41598-018-33476-x
Does this algorithm distinguish promotors and enhances (in upstream/downstream sequences)?
Comments on the Quality of English LanguageMinor editing of the English language required
Author Response
Reviewer 3
- Title: How does the author name the algorithm "BestCRM", if in future some better algorithm will be developed then how this "Best" will be justified? The name of the algorithm should be changed.
This is exactly the idea behind the exhaustive search. By the exhaustive search all combinations are examined. In most situations this takes enormous amount of time, therefore simplifications are applied to find sub optimal results but quicker. That's why newer methods with different heuristics may find better results. Exhaustive search, or checking all variants of CRMs, guarantees that there are no more optimal CRM for the given data set.
- Graphical unit interface or URL links should be given to show how this algorithm looks and works.
This program is intended to be included in pipelines and used iteratively, and mostly under Linux environment. That is why it has a command line interface. The program is distributed for local use by bioinformatitions. As an example we may suggest the following program, which is also have only command line interface:
Yi R, Cho K, Bonneau R. NetTIME: a multitask and base-pair resolution framework for improved transcription factor binding site prediction. Bioinformatics. 2022 Oct 14;38(20):4762-4770. doi: 10.1093/bioinformatics/btac569. PMID: 35997560; PMCID: PMC9563695.
NetTIME is freely available at https://github.com/ryi06/NetTIME
- Picture of the BestCRM should be given as a Figure or links to allow users to interact with it.
Link is given in Data Availability Statement: The program source code, executables for Windows and Linux, to-gether with all required libraries and examples are freely available at https://github.com/Deyneko/BestCRM.
- Introduction: Providing some examples of related computational algorithms with online links will be useful for the reader.
Now several programs for short motif and CRM discovery added, the reader should follow citated publications, since not all programs have GUI.
- Figure 1: The graphics of the figure are not clear, and no legends are visible.
I'm very sorry for that. As a reviewer I also receive sometimes documents with images in low quality. But now I know what is the problem - the software on mdpi website reduces the quality too strong when converting from docx to PDF. Now I have converted myself retaining original quality. Ps. Fig 1 are original image produced by evaluation software, I would like to keep it original as is.
- From where the sequence data was collected to test the algorithm?
It is from published datasets, both for known CRMs and new CRMs.
- How does the author say the algorithm works best only based on identifying the promotor with minimum input parameters? More input parameters will usually give the best output.
Required parameters are calculated using the promoter sets. More input parameters will not always give better results – correct values for most of the parameters are unknown and user may only guess what they can be.
- Is BestCRM only used for Humans? Can this method identify the CRM of prokaryotes (bacteria/archaea)? If it is, then how is it distinguished from different eukaryotes? Does BestCRM reveal the specific patterns in different Extremophiles or Archaea? Then, cite the following paper: https://www.nature.com/articles/s41598-018-33476-x
It can be used for any organism. Program is supplied with PWMs for bacteria, it will not necessarily find differences in Extremophiles or Archaea, but can be applied for analysis. Citation added.
- Does this algorithm distinguish promotors and enhances (in upstream/downstream sequences)?
The algorithm is used to analyze promoters and enhancers to find common elements they (may) have. Recognition is a different task.
Round 2
Reviewer 1 Report
Comments and Suggestions for Authors
Author has addressed majority of my comments however few minor modifications are still required for the following comments.
My earlier comment:- Mention the performance metrics in the abstract
Authors answer: Added “… methods according several metrics like specificity, sensitivity, AUC etc.”
For this comment add the values of your performance metrics in the abstract.
My earlier comment:- Line no 100-what are the other statistical methods, name them
Authors answer: These are on figure 1, now in better resolution.
I don't find the names in Figure 1 rather that are in Figure 2. I meant to say that you should write the names of the methods in the text as "other statistical methods like, (names of the methods)" .
Comments on the Quality of English Language
Minor English editing is still required.
Author Response
1) Mention the performance metrics in the abstract. Authors answer: Added “… methods according several metrics like specificity, sensitivity, AUC etc.” For this comment add the values of your performance metrics in the abstract.
R1) Values of the specificity, sensitivity, AUC etc, have only meaning when compared to same values of other methods. These values strongly depend from datasets and many other settings, therefore giving values for one method has no sense. For example, 0.63 AUC is a good or bad performance? For some methods it is perfect, for other it is unacceptably low. I would need to copy most of the results into abstract to show my values are higher than from other methods.
No one is giving performance values in abstract:
- PC-TraFF: identification of potentially collaborating transcription factors using pointwise mutual information Cornelia Meckbach1*, Rebecca Tacke1, Xu Hua1, Stephan Waack2, Edgar Wingender1 and Mehmet Gültas1*
- Identification of functional clusters of transcription factor binding motifs in genome sequences: the MSCAN algorithm O. Johansson1,W.Alkema2,W.W.Wasserman3,∗ and J. Lagergren4
2) My earlier comment:- Line no 100-what are the other statistical methods, name them. Authors answer: These are on figure 1, now in better resolution. I don't find the names in Figure 1 rather that are in Figure 2. I meant to say that you should write the names of the methods in the text as "other statistical methods like, (names of the methods)" .
R2) Another reviewer asked for method workflow, so new fig 1 was introduced, and the old fig 1 became fig 2. I inserted all methods into the text. Line 120.
The English is checked again.
Reviewer 2 Report
Comments and Suggestions for Authors
Thank Igor V. Deyneko for addressing the concerns raised during the initial review. I appreciate the thorough revisions made to the manuscript. The structure of the manuscript was significantly improved after the revision. With the implemented revisions, I believe the manuscript is now suitable for publication. Thank you again for your diligence in addressing the feedback. I look forward to seeing more of your work in the future.
Author Response
Thank you
Reviewer 3 Report
Comments and Suggestions for Authors
No more comments. The author responded well to the queries.
Author Response
Thank you